# Individuals co-exposed to sand fly saliva and filarial parasites exhibit altered monocyte function

Moussa Sangare[1,2]*, Yaya Ibrahim Coulibaly[1,3], Naureen Huda[4], Silvia Vidal[5], Sameha Tariq[6], Michel Emmanuel Coulibaly[1], Siaka Yamoussa Coulibaly[1], Lamine Soumaoro[1], Ilo Dicko[1], Bourama Traore[1,3], Ibrahim Moussa Sissoko[1], Sekou Fantamady Traore[1], Ousmane Faye[1,3], Thomas B. Nutman[6], Jesus G. Valenzuela[7], Fabiano Oliveira[7], Seydou Doumbia[1], Shaden Kamhawi[7], Roshanak Tolouei Semnani[8]*

1 Mali International Center for Excellence in Research, Faculty of Medicine and Odonto-Stomatology, University of Sciences, Techniques and Technologies of Bamako, Bamako, Mali, 2 Interdisciplinary School of Health Sciences, Faculty of Health Sciences, University of Ottawa, Ottawa, Canada, 3 Dermatology Hospital of Bamako, Bamako, Mali, 4 Department of Pediatrics, University of California, San Francisco, California, United States of America, 5 Institut Recerca H. Sant Pau C. Sant Quintí, Spain, 6 Laboratory of Parasitic Diseases, LPD, National Institute of Allergy and Infectious Diseases, National Institutes of Health, Bethesda, Maryland, United States of America, 7 Vector Molecular Biology Section, LMVR, National Institute of Allergy and Infectious Diseases, National Institutes of Health, Rockville, Maryland, United States of America, 8 Autoimmunity and Translational Immunology, Precigen, Inc. *A wholly owned subsidiary of Intrexon Corporation*, Germantown, Maryland, United States of America

* mbsangare@icermali.org (MS); rsemnani@precigen.com (RTS)

**Data Availability Statement:** All relevant data are within the manuscript and its Supporting information files.

## Abstract

### Background

In Mali, cutaneous leishmaniasis (CL) and filariasis are co-endemic. Previous studies in animal models of infection have shown that sand fly saliva enhance infectivity of *Leishmania* parasites in naïve hosts while saliva-specific adaptive immune responses may protect against cutaneous and visceral leishmaniasis. In contrast, the human immune response to *Phlebotomus duboscqi* (Pd) saliva, the principal sand fly vector in Mali, was found to be dichotomously polarized with some individuals having a Th1-dominated response and others having a Th2-biased response. We hypothesized that co-infection with filarial parasites may be an underlying factor that modulates the immune response to Pd saliva in endemic regions.

### Methodology/Principal findings

To understand which cell types may be responsible for polarizing human responses to sand fly saliva, we investigated the effect of salivary glands (SG) of Pd on human monocytes. To this end, elutriated monocytes were cultured in vitro, alone, or with SG, microfilariae antigen (MF ag) of *Brugia malayi*, or LPS, a positive control. The mRNA expression of genes involved in inflammatory or regulatory responses was then measured as were cytokines and chemokines associated with these responses. Monocytes of individuals who were not exposed to sand fly bites (mainly North American controls) significantly upregulated the

**Funding:** This work was supported by the National Institutes of Health under Award Number P50AI098505 (Tropical Medicine Research Center, TMRC) through the Division of extramural Research Program of NIAID, and the Fogarty International Center (grant ID D43TW008652) of the National Institutes of Health, Bethesda, MD, USA (https://www.nih.gov/). This research was partly supported by the Intramural Research Program of the NIH, National Institute of Allergy and Infectious Diseases. The following authors (MS, YIC, ID, MIS and BT) received salaries from the funder (TMRC). The grant was awarded to SD at the University of Bamako. The funders had no role in study design, data collection and analysis, decision to publish, or preparation of the manuscript.

**Competing interests:** The authors have declared that no competing interests exist.

production of IL-6 and CCL4; cytokines that enhance *leishmania* parasite establishment, in response to SG from Pd or other vector species. This selective inflammatory response was lost in individuals that were exposed to sand fly bites which was not changed by co-infection with filarial parasites. Furthermore, infection with filarial parasites resulted in upregulation of CCL22, a type-2 associated chemokine, both at the mRNA levels and by its observed effect on the frequency of recruited monocytes.

## Conclusions/Significance

Together, our data suggest that SG or recombinant salivary proteins from Pd alter human monocyte function by upregulating selective inflammatory cytokines.

### Author summary

In Mali, cutaneous leishmaniasis (CL) and filariasis are co-endemic. We hypothesized that co-infection with helminth parasites may modulate the immune response to saliva of the CL vector *Phlebotomus duboscqi* (Pd). Hence, we investigated the effect of Pd salivary glands (SG) on human monocytes in subjects exposed or unexposed to sand fly bites in the context of a concomitant filaria infection. Monocytes of unexposed individuals selectively upregulated the production of IL-6 and CCL4 in response to SG from Pd or other vector species. In contrast, monocytes of individuals exposed to sand fly bites lost their responsiveness to SG, microfilariae antigen and LPS, irrespective of co-infection with filaria. Nevertheless, infection with filaria significantly upreguled the frequency of CCL22+monocytes, IL-10+mDCs and regulatory T cells. Together, our data suggest that repeated exposure to Pd saliva alters human monocyte function towards a tolerized phenotype while co-infection with filaria favors a *Leishmania*-promoting Th2/regulatory response.

## Introduction

Cutaneous leishmaniasis, a vector-borne disease transmitted by bites of infected phlebotomine sand flies [1], is endemic in sub Saharan Africa including Mali [2–4]. Sand flies are found in areas endemic for lymphatic filariasis (LF) that results in physical disabilities and carries an important socioeconomic impact [5]. In Mali, the major filarial parasite responsible for LF is *Wuchereria bancrofti* (Wb) but *Mansonella perstans* (Mp) is also present [6].

Sand fly saliva has an array of proteins with varied pharmacological properties that facilitate blood feeding [7]. Salivary proteins also exert immunomodulatory effects on the innate immune system [8–11] and induce a strong adaptive immunity [12–14]. Vector-transmission of *Leishmania* parasites is exclusively by the bite of infected sand flies where the parasites are co-deposited into the skin together with saliva. Studies have consistently shown that saliva generally enhances infectivity of *Leishmania* parasites in a naïve host [8] while an adaptive immune response to whole saliva or a distinct salivary protein generally protects against cutaneous and visceral leishmaniasis in rodent models of infection [15–19]. However, for certain sand fly species and certain salivary proteins induction of a specific adaptive immune response exacerbates *Leishmania* infection [19,20]. Protective salivary proteins are characterized by their induction of a Th1 delayed-type hypersensitivity response where IFN-γ, considered a

strong correlate of protection in leishmaniasis, is induced. The mechanism of this protection remains to be fully elucidated but preliminary evidence points to priming of a protective anti-*Leishmania* response at the site of bite mediated by immunity to saliva [21,22]. In contrast, disease exacerbation is correlated with the induction of Th2 and/or regulatory cytokines [19]. Overall, such findings suggest that salivary proteins are potent modulators of the host immune response with consequences for the transmitted *Leishmania* parasites. While the majority of these studies were undertaken in animal models, the human immune response to saliva of *Phlebotomus duboscqi* (Pd), the principal sand fly vector in Mali, was found to be dichotomously polarized with some individuals developing a Th1-dominated response and others a Th2-biased response [23].

The altered function of the innate immune response, particularly the dysfunction of antigen presenting cells (APC), has been postulated to mediate some of the parasite-specific T cell unresponsiveness seen in patent filarial infection. Previously, we have shown that live MF of *Brugia malayi* (Bm) modulate dendritic cell (DC) function by altering the expression and function of Toll-like receptors 3 and 4, and by inducing apoptotic DC cell death [22]. In addition to the effect of live MF on DCs, we have shown that circulating monocytes from filaria-infected individuals are laden with filarial antigens, exhibit diminished antigen presentation and processing, and produce fewer proinflammatory cytokines in response to surface receptor cross-linking [24]. Therefore, we hypothesized that a filarial infection may compromise the immune response to sand fly salivary proteins, which in turn may alter the host's immune response to *Leishmania* infection. In this study, we aimed to assess the effect of sand fly saliva on human monocytes in the context of a concomitant filarial infection.

## Methods

### Ethics statement

The Institutional Review Board of the National Institutes of Health, USA, and the Ethical Committee of the Faculty of Medicine, Pharmacy and Odonto-Stomatology of Bamako, Mali approved the study (FWA 00001769, IRB00001983). Leucopacks from health blood bank donors were obtained from the Department of Transfusion Medicine (National Institutes of Health, Bethesda, MD) under (IRB)-approved protocols (IRB# 99-CC-0168). Written informed consent was obtained from all the volunteers prior to their participation.

### Study populations

We screened 930 individuals in the Southern districts of Mali, Kolondieba (village of Boundioba) and Kolokani (villages of Tieneguebougou and Bougoudiana). These villages are co-endemic for filaria and leishmania infections [25,26].

The individuals were screened for *Leishmania* using the leishmanin skin test (LST); for filarial infections using the Immunochromatographic card test (ICT) for LF and thick blood smear for *Mansonella perstans* (Mp); and for antibodies to sand fly saliva by ELISA. All screened individuals were determined to be positive by sand fly antibody ELISA (SF+) and negative by LST test. Among those, we identified 67 SF+ and filaria or microfilaria-infected clinically asymptomatic (MF+) individuals and 61 SF+/MF- individuals.

### Thick blood smear

Finger-prick and venous blood samples were collected during daytime using a calibrated thick blood smear (60μl). The blood was drawn onto a microscope slide, dried and stained with 10%

Giemsa using standard procedures. The stained smears were examined using a light micro-scope with a 10× objective for the detection of Mp microfilariae.

### ICT

The ICT Filariasis test is a rapid-format filarial antigen test that was developed by ICT diagnostics (Filariasis test, Alere Inc., Scarborough, USA). The test is designed for detection of soluble *W*. *bancrofti* antigens that circulate in the blood of infected humans.

### LST

LST material is supplied as sterile 2.5ml aqueous suspensions, with each ml containing $6 \times 10^6$ killed *L*. *major* promastigotes (strain MRHO/IR/75/ER) in phosphate buffered saline and 0.01% thimerosal, pH 7.0–7.1. Leishmanin (0.1ml) was injected intra-dermally in the left fore-arm. Readings were taken 48 to 72 hours after injection using a ball point pen to determine the induration size. Measurements with a diameter greater than 5mm were considered positive [12].

### Anti-saliva ELISA

The blood sample made from the filter paper was obtained using a 6 mm disposable punch, eluted in 0.05% Tween PBS, and stored at room temperature (RT) overnight. The serum obtained from this process is referred to as "eluted blood". The ELISA test was carried out as described previously [26]. The ELISA plates were coated with 50μl of Pd salivary gland homogenate diluted at 2μg/ml in carbonate/bicarbonate buffer (pH 9.6) overnight at 4˚C. The next day the plates were washed three times with PBS- 0.05% Tween, and blocked with PBS containing 4% bovine serum albumin for 2 hours at room temperature. The plates were washed and sera (100 μl of sera (diluted 1:100 in PBS-Tween+0.5% BSA)) were added and incubated at RT for one hour. After washing, alkaline phosphatase conjugated goat anti-human IgG (Sigma, MO) was added to each well (100μl/well (1:1000 in PBS-Tween-BSA)) and incubated at 37˚C for an hour. Following another washing, *p-nitrophenyl phosphate* substrate (Sigma Sr. Louis, MO) was added and the absorbance was read at 405nm on a Versamax microplate reader after 30 minutes. Values obtained were subtracted from those obtained for the background (where PBS replaced "eluted blood").

### MF antigen preparation

Soluble MF Ag was made from $\sim 10^8$ live *Brugia malayi* MF (provided by John McCall, University of Georgia, Athens, Ga.) as described previously [27]. The MF were washed repeatedly in RPMI medium containing antibiotics and cultured overnight at 37˚C in 5% CO2. Worms were harvested the following day, washed with PBS and frozen at −20˚C. The frozen MF were pulverized, sonicated, and extracted in PBS at 37˚C for 4 hours and then at 4˚C overnight. Following centrifugation at 20000g for 30 minutes, the supernatant was passed through a 0.45μm filter and stored in aliquots at −70˚C. The antigen was found to be endotoxin free by QCL-1000 kit.

### Sand flies and preparation of SG homogenate

Sand flies were reared at the Laboratory of Malaria and Vector Research, NIAID, NIH or were kindly provided by Walter Reed Army Institute of Research. SGs were dissected from 5- to 7-day-old females and stored in 20 μl phosphate-buffered saline at -70C. Salivary glands were

sonicated followed by centrifugation at 12,000 x g for 3 minutes. Supernatants were collected and stored at -80 C until use.

## Recombinant proteins preparation

The sand fly salivary proteins were chosen based on their observed immunogenicity in humans [28–30]. The recombinant protein 1H2 (Pdum34; accession no. DQ834330) is a PpSP32-like silk-related protein, while 4B4 (Pdum35; accession no. DQ826522) and 4C2 (Pdum10; accession no. DQ826519) belong the yellow family of proteins from Pd saliva [31]. The function of silk-related proteins remains unknown but yellow proteins have biogenic amine binding properties [32,33]. Additionally, 4B4 and 4C2 have been recently shown to act as neutrophil chemoattractants [30]. LJL143 (accession no. AY445936) from saliva of *Lutzomyia longipalpis* has both anticoagulant [34] and anti-complement [35] activities. Briefly, cDNA encoding for each of the target salivary proteins from Pd and *Lutzomyia. longipalpis* (Ll) was cloned into the VR2001-TOPO vector, expressed in HEK-293F cells, and purified using HPLC as previously described [36].

*Leishmania* antigen, kindly donated by Dr. David L. Sacks (NIAID/NIH) was prepared by freeze/thaw cycles of *L. major* stationary-phase promastigotes as described previously [37].

## In vitro culture of monocytes from North American individuals

Leucopacks from healthy blood bank donors were obtained from the Department of Transfusion Medicine (see ethics statement above). Human monocytes were cultured at $50 \times 10^6$ per 6-well plate in serum-free medium for 2 hours, after which the medium was removed and complete medium (RPMI 1640 medium, Thermo Fisher Scientific, MA. supplemented with 10% heat-inactivated fetal calf serum, Gemini Bioproducts, Sacramento, CA 20 mM glutamine, 100IU/ml penicillin, and 100g/ml streptomycin, Biofluids, Inc., Rockville, MD) was added.

## In vitro activation of monocytes from North American individuals

Human monocytes were harvested and further cultured at $1 \times 10^6$ cells/ml either in media alone, or with LPS (1μg/ml), MF antigen (10μg/ml), salivary glands from different sand fly species (1 salivary gland pair/$1 \times 10^6$ cells), salivary recombinant proteins 1H2, 4B4 and 9G11 from Pd or LJL143 from Ll (2.5μg/ml), or *Leishmania* antigen (10μg/ml) for 30 minutes (mRNA expression was measure by qPCR) and 24 hours (cytokine production was measured by Luminex). At the indicated time points, cells were harvested to measure mRNA expression, and the supernatants were collected to assess cytokine and chemokine production.

## Monocyte isolation from whole blood of MF$^+$/SF$^+$ and MF$^-$/SF$^+$ individuals

Peripheral blood mononuclear cells (PBMC) from the blood of Malian volunteers or blood bank donors were isolated by Ficoll diatrizoate density centrifugation [38]. Monocytes were isolated from PBMC using a CD14-negative monocyte isolation kit (Dynal Biotech, Lake Success, NY). Typically, the monocytes were ≥96% pure as assessed by flow cytometry.

## Monocyte activation from MF$^+$/SF$^+$ and MF$^+$/SF$^-$ individuals

Monocytes isolated using CD14-negative isolation kit (Dynal Biotech) were cultured at $1 \times 10^6$ cells/ml either in media alone, or with LPS (1μg/ml), MF antigen (10μg/ml), salivary glands from different sand fly species (1 salivary gland pair/$1 \times 10^6$ cells), *Leishmania* antigen (10μg/ml) for 24 hours. Cells were harvested and the supernatants were collected to assess cytokine and chemokine production.

## RNA preparation and RT-PCR

RNA was extracted using the RNeasy kit (Qiagen, Valencia, CA) according to the manufacturer's instructions. cDNA generation was performed using random hexamers and reverse transcriptase (ABI, Grand Island, NY). Expression of genes (inflammatory: IP10, IL-6, CCL2, CCL4, CCL3, PDL1, and IDO; type-2 associated: CCL22; regulatory: TGF-β, IL-10) were determined by quantitative PCR using TaqMan reagents (ABI) on an ABI 7900HT. The threshold cycles (CT) for 18S and genes of interest were calculated and used to determine the relative transcript levels. The formula 1/delta CT was used to determine the relative transcript levels, where delta CT is the difference between the CT of the target gene and the CT of the corresponding endogenous 18S reference gene.

## Cytokine analysis

The concentration of cytokines in culture supernatants was assessed using the Milliplex human cytokine/chemokine magnetic bead panel kit and the Luminex 100/200 system (Luminex Corporation, Austin, TX, USA). The lower limit of detection was 3.2pg/mL for IL-1α, IL-1β, IL-6, CCL4, IP10, CCL2, IL-10, IL-5, IL-17, and IFN-γ. Supernatant cytokine concentrations were determined by ELISA for CCL22 at a sensitivity of 12.5pg/mL according to the manufacturer's protocol.

## Whole blood cell activation

Whole blood was collected from SF+/MF- and SF+/MF+ individuals and diluted with serum-free media at a 1:1 ratio and cultured in 1ml per condition: media alone, with LPS (1μg/ml), MF ag (20μg/ml), Leish ag (20μg/ml) or SG Pd (1 pair/ml) for 24 hours. Supernatants were collected and cytokine concentrations were measured using Luminex or ELISA. In parallel wells, 12 hours after stimulation, BFA (10μg/condition) was added to permeabilize cells to measure intracellular cytokines. After 24 hours, red blood cells were lysed using 1x BD lysing solution (BD Biosciences, CA), fixed and labeled with antibodies for flow cytometry.

## Flow cytometery: Staining and gating strategy

Cryopreserved whole blood was thawed at room temperature, washed with RPMI 1640 plus 5mm EDTA (Life Technologies, Inc.), and diluted in RPMI 1640 supplemented with 10% FCS. Viability was determined using CD45 FITC (BD Biosciences Becton Dickinson). Cell pellets (100μl) were incubated with different panels of antibodies including anti-CD3-Alexa Fluor700, anti-CD4-APC, anti-CD127-PE-Cy5 (Bio Legend), anti-CD25-APC-Cy7, anti-CD14-APC-Cy7, anti-CD15-PE-Cy5, anti-CD19-PE-Cy7, anti-CD11c PE-Cy5, anti-HLA-DR-APC-Cy7, anti-CD123-PE-Cy7 (BD Biosciences) and anti-CD16-Pacific Orange (Invitrogen). In those panels to determine the production of chemokines, we also include anti-CCL22-PE and anti-IL-10- Pacific Blue (BD Biosciences). After 20 minutes of staining, cells were fixed using a BD FACS lysing solution (BD Biosciences). Cells were analyzed on a Fortessa flow cytometer.

Tregs were analyzed gating on CD3+ CD4+ CD25+ CD127low and mDCs were analyzed gating on Lineage (CD3, CD14, CD20 FITC) negative HLA-DR+CD11c+CD123+

## Statistical analysis

Statistical analysis of data from real-time PCR, Luminex, and ELISA was done using a Wilcoxon matched pairs signed-rank test, and for clinical data the nonparametric Mann-Whitney

test was performed to compare between SF⁺/MF⁺ and SF⁺/MF⁻ individuals using Graph Pad Prism version 8 (GraphPad Software, Inc.).

## Results

### Saliva of the vector sand fly *Phlebotomus duboscqi* (Pd) upregulates the production of CCL4 and IL-6 in human monocytes

Recruitment and immunomodulation of monocytes at the site of a sand fly bite following exposure to saliva has been reported in the past [39]. Here, we assessed the effect of SG Pd on polarization of monocytes from healthy donors and compared it to the effect of MF ag, *Leishmania* antigen (Leish ag), or LPS (positive control). To assess the phenotype of monocytes, we measured the production of cytokines associated with a proinflammatory response (IL-6, IP10, CCL2, CCL3, CCL4), a type-2 response (CCL22), and a regulatory response (IL-10). Production of the proinflammatory cytokines IL-6 and CCL4 (MIP-1-β) was significantly upregulated after stimulation for 24 hours with either SG Pd, MF ag or Leish ag (Fig 1). Interestingly, SG Pd had the most restrictive effect on monocytes, only upregulating IL-6 and CCL4. In

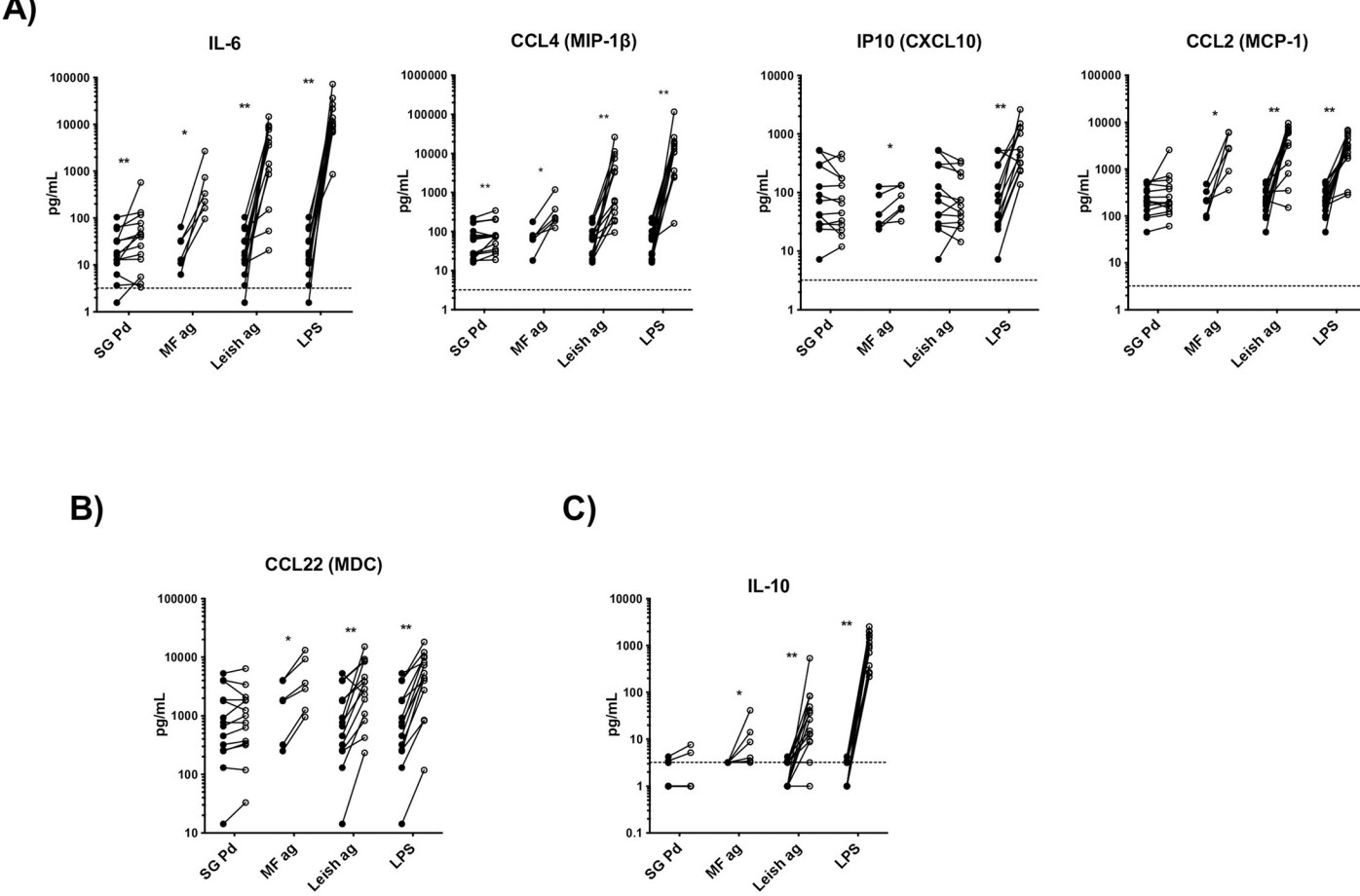

**Fig 1. Saliva of the vector sand fly *Phlebotomus duboscqi* upregulates the production of CCL4 and IL-6 in human monocytes.** Human monocytes were unexposed (closed circles) or exposed (open circles) to either saliva of SG Pd, MF ag, Leish ag, or to LPS (positive control) for 24 hours *in vitro*. Supernatants were collected and the levels of A) Type-1 associated chemokines/cytokines IL-6, CCL4, IP10, and CCL2, B) Type-2 associated chemokine CCL22, and C) regulatory cytokine IL-10 were measured. Each line represents an independent donor (*n* = 6–14). Statistical difference in cytokine production between unexposed and exposed cells are measured using Wilcoxon matched pairs signed-rank test; * (*p*<0.05); ** (*P*<0.005).

comparison, MF ag upregulated monocytes production of all the tested cytokines (Fig 1). Leish ag also upregulated the production of all cytokines with the exception of IP10. Of note, despite the pronounced response at the protein level, mRNA expression of selected genes was not altered after 4 hours of stimulation with SG Pd, Pd recombinant salivary proteins 1H2 and 4B4, or MF ag (S1 Fig). The difference could be due to the shorter exposure to stimuli to evaluate gene expression. Further, only the mRNA levels for CCL2 and CCL4 were significantly induced by Leish ag (S1 Fig).

## Saliva of several vector species of sand flies upregulates the production of CCL4 and IL-6 in human monocytes

As some proteins in saliva of sand fly species are species-specific, and most have been shown to possess immunomodulatory effects on T cells, macrophages and DC populations, we next examined the effect of saliva of several sand fly species on the modulation of human monocytes from healthy donors. Similar to SG Pd, saliva of *Phlebotomus papatasi* (SG Pp), *Phelobotomus argentipes* (SG Pa), *Phleobotomus sergenti* (SG Ps), *Phelobotomus perniciosus* (SG Ppr), and *Lutzomyia longipalpis* (SG Ll) significantly upregulated the production of CCL4 and IL-6 in human monocytes (Fig 2). Additionally, CCL2 production was also significantly upregulated by SG Ll and SG Ps, while the production of a type 2 cytokine CCL22 was only enhanced by Ppr demonstrating global and species-specific effects of sand fly saliva on human monocytes (Fig 2).

## Immunomodulatory effects of salivary and parasitic antigens are lost in monocytes of individuals naturally exposed to vector bites

Next, we aimed to assess the response of monocytes from individuals that are naturally exposed to saliva through vector bites in the context of a helminth infection. We tested individuals in regions of Mali co-endemic for both Pd sand flies and filarial infection. The first group are individuals who are sand fly bite positive (SF+) and MF negative (MF-), hereafter referred to as SF+/MF-. The second group are individuals positive for sand fly bite (SF+) and positive for MF (MF+), hereafter referred to as SF+/MF+ (see Study populations section for a full description). To assess cytokine production, monocytes of both groups were isolated from PBMC and cultured overnight in either media alone or with SG Pd, MF ag, or LPS (positive control).

Interestingly, stimulation with SG Pd did not induce any of the proinflammatory, type-2 or regulatory cytokines in monocytes of both groups despite having originated from individuals naturally exposed to sand fly bites (Fig 3). In fact, while IL-6 and CCL4 were upregulated in monocytes from sand fly bite-naive individuals (Fig 2), this response was lost in SF+ individuals regardless of their MF status (Fig 3). More importantly, exposure to sand fly bites also dampened the monocyte response to MF antigen and even to LPS, observed for some cytokines (Fig 3). Supporting previous data [24,40], monocytes of MF+ individuals were less responsive than monocytes of MF- individuals since LPS significantly induced the production of CCL4 and IL-10 only in MF- individuals (Fig 3). Of note, no significant differences between the SF+/MF- and SF+/MF+ individuals were observed when basal cytokine production (media alone) was measured (S2 Fig). Collectively, these data suggest that monocytes from individuals repeatedly exposed to sand fly bites exhibit a dampened response to stimulation with any antigen. This could be due to a much higher basal cytokine expression in monocytes of individuals naturally exposed to sand fly bite (Fig 3; closed circles) than unexposed individuals from North America (Fig 1; closed circles).

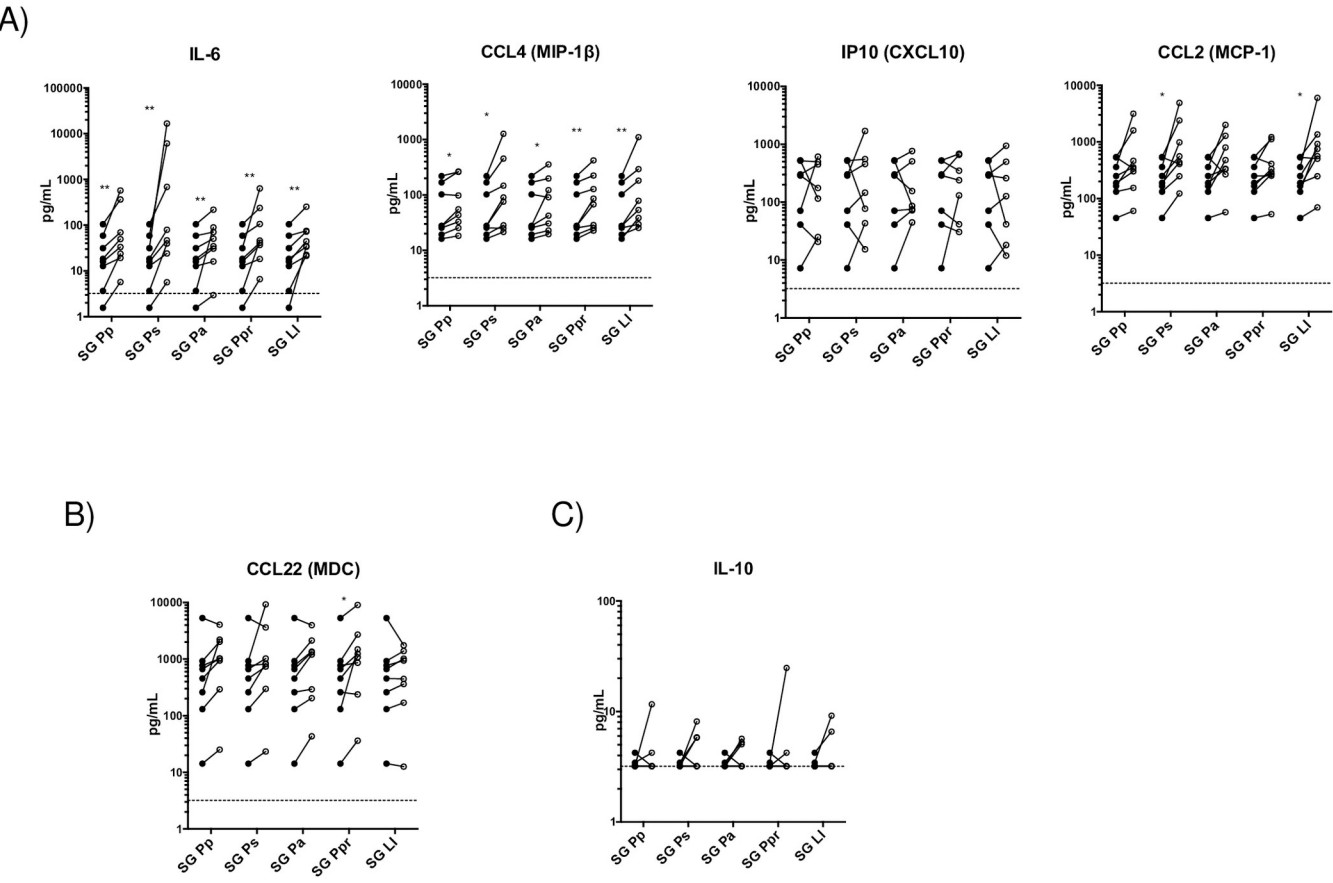

**Fig 2. Saliva of several vector species of sand flies upregulates the production of CCL4 and IL-6 in human monocytes.** Human monocytes were unexposed (closed circles) or exposed (open circles) to saliva of either SG Pp, SG Pa, SG Ps, SG Ppr, and SG Ll for 24 hours. Supernatants were collected and the levels of A) Type 1-associated chemokines/cytokines IL-6, CCL4, IP10, and CCL2, B) Type-2 associated chemokine CCL22, and C) regulatory cytokine IL-10 were measured. Each line represents an independent donor ($n = 6–8$). Statistical difference in cytokine production between unexposed and exposed cells are measured using Wilcoxon matched pairs signed-rank test; * ($p<0.05$); ** ($P<0.005$).

### Increase in mRNA expression of CCL22 and frequency of CCL22⁺ monocytes in SF⁺/MF⁺ than SF⁺/MF⁻ individuals

To assess the effect of helminth infection on gene expression, we measured the basal mRNA expression in monocytes of SF⁺/MF⁻ compared to SF⁺/MF⁺ individuals. While there was no difference in proinflammatory and regulatory cytokine production, monocytes of SF⁺/MF⁺ individuals showed a significantly higher expression of CCL22, a type-2 chemokine (Fig 4), suggesting that exposure to sand fly saliva does not change the MF-induced type-2 phenotype.

Worth to mention, whole blood collected from SF⁺/MF⁺ and SF⁺/MF⁻ individuals, cultured in media alone showed no difference in the production of any of the inflammatory, type-2 or regulatory cytokines between the groups (S3 Fig).

### SF⁺/MF⁺ individuals have a higher frequency of Treg and IL-10⁺ mDC than SF⁺/MF⁻ individuals

To understand whether exposure to sand fly bite changes the composition of innate immune cells, we performed flow cytometry on whole blood of SF⁺/MF⁻ and SF⁺/MF⁺ individuals. To this end, we did not see a significant difference in the frequency of lymphocytes or neutrophils,

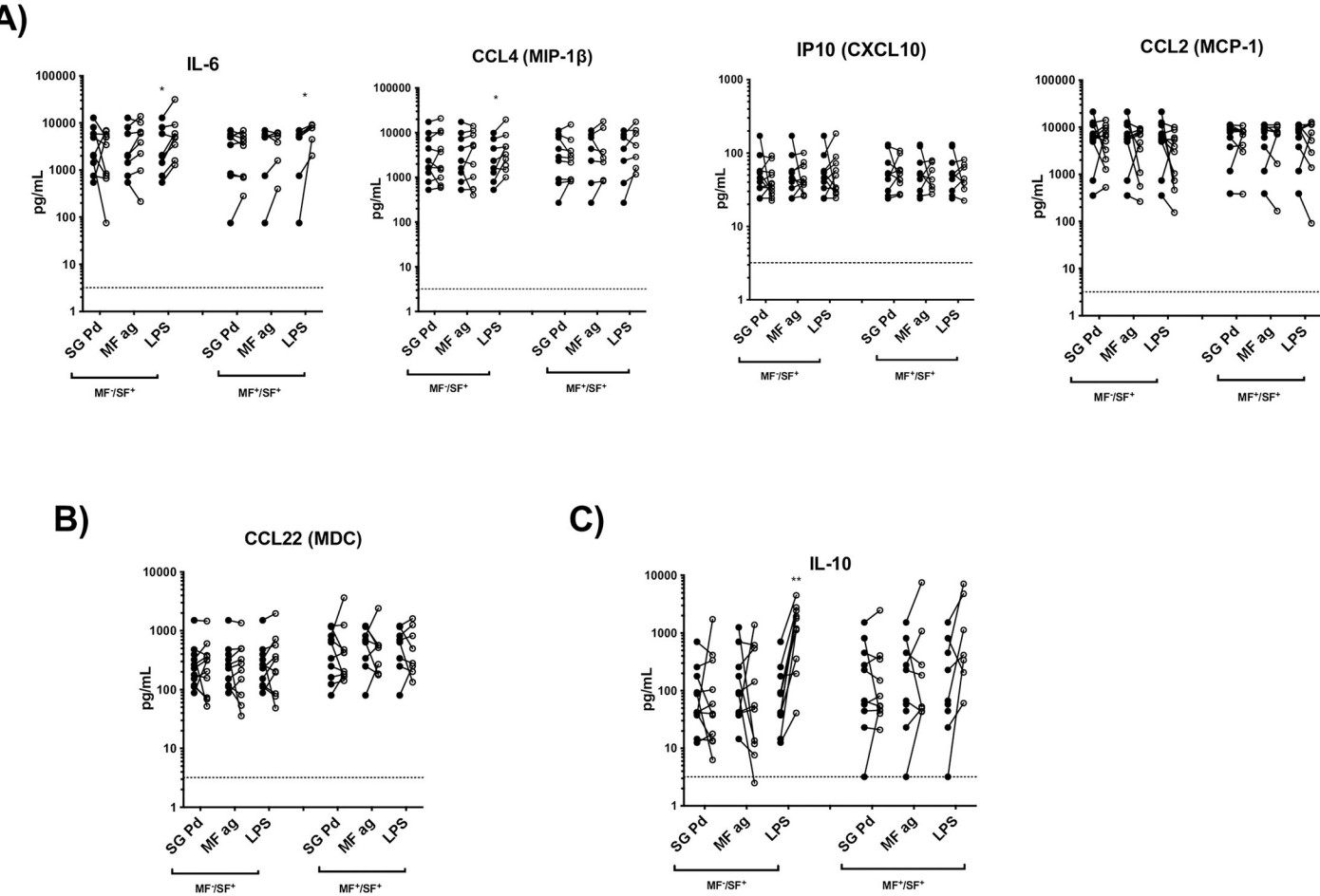

**Fig 3. Upregulation of CCL4, CCL2 and IL-6 production by saliva of the sand fly *Phlebotomus duboscqi* is lost in monocytes of individuals naturally exposed to vector bites.** Monocytes from PBMCs of SF$^+$/MF$^+$ (n = 11), and SF$^+$/MF$^-$ (n = 10) individuals were isolated and cultured in media alone (closed circles) or exposed to (open circles) either SG Pd, MF ag, or LPS for 24 hours. Supernatants were collected and cytokine levels were measured by Luminex. Each line represents an individual. Statistical difference in cytokine production between unexposed and exposed cells are measured using Wilcoxon matched pairs signed-rank test; * ($p<0.05$); ** ($P<0.005$).

but observed a significant increase in the percentage of CD25$^+$/CD127$^{low}$ Tregs in SF$^+$/MF$^+$ individuals (Fig 5A). Further, though there was no overall difference in the number of monocytes or mDC in SF$^+$/MF$^+$ as compared to SF$^+$/MF$^-$ individuals, a significant increase was observed in the frequency of CCL22$^+$ monocytes and IL-10$^+$ mDCs in SF$^+$/MF$^+$ as compared to SF$^+$/MF$^-$ individuals (Fig 5). These data suggest that exposure to sand fly saliva through bites doesn't change the regulatory phenotype that has already been established by helminth infection.

## Discussion/Conclusion

To date, most studies of sand fly saliva have been focused on leishmaniasis; understandable as sand flies are biological vectors that transmit *Leishmania* by bite. Nevertheless, in real-life situations humans are exposed to multiple infections. Importantly, co-infections have been shown to shape the immune response to pathogens including helminths, *Plasmodium* and *Leishmania* [41–43]. Another underexplored yet equally relevant facet to immunomodulation is that humans are frequently exposed to saliva of biting disease vectors that contains

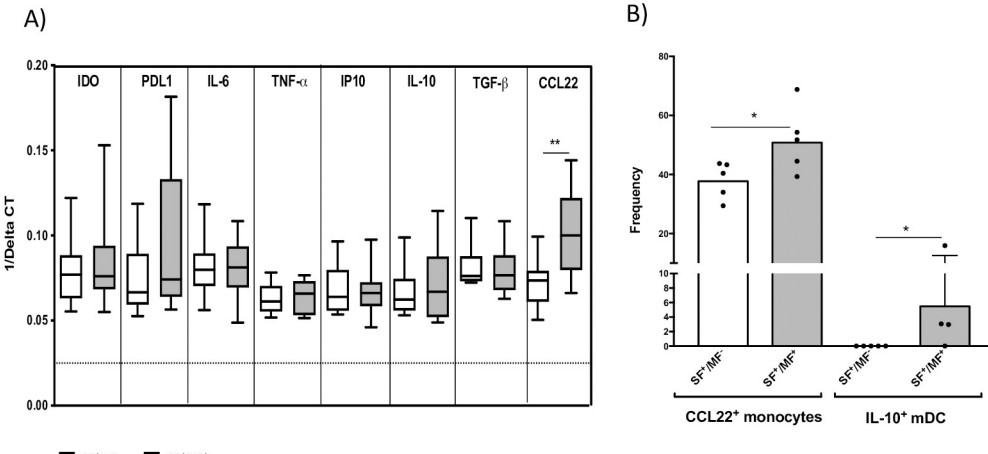

**Fig 4. mRNA expression of CCL22 and the frequency of CCL22+ monocytes and IL-10+ mDC is higher in SF+/MF+ than SF+/MF− individuals.** A) Monocytes from PBMCs of SF+/MF+ (n = 11) and SF+/MF− (n = 10) individuals were isolated and cultured in media for 24 hours. Cells were harvested, and A) mRNA levels were measured by TaqMan real-time PCR and normalized to the levels of 18S rRNA. Results are shown as geometric mean of 1/average ΔCT and graphed as boxes and whiskers (with minimum and maximum; there is no mRNA detection below 0.0277 (the dotted line). B) Whole blood from SF+/MF+ (n = 5) and SF+/MF− (n = 5) individuals were RBC-lysed and cultured in media alone for 24 hours. Cells were harvested and the frequency of intracellular CCL22+ cells in CD14+/HLA-DR+ monocytes IL-10+ cells in HLA-DR+/CD11c+/CD123− mDCs were measured by flow cytometry. Bars represent geometric mean of frequency in SF+/MF+ (n = 5) and SF+/MF− (n = 5). Statistical difference in between the two groups are measured using nonparametric Mann- Whitney test; * ($p<0.05$).

immunomodulatory and immunogenic proteins [44–46]. It has been established that sand fly saliva contains immunomodulatory molecules that manipulate the host immune response exacerbating infection by *Leishmania*, while adaptive immunity generated against salivary proteins in experimentally- or naturally-exposed hosts associates to protection from leishmaniasis [44].

Filariasis and leishmaniasis are co-endemic in Mali where the majority of inhabitants are also exposed to sand fly bites [25]. Here, we aimed at understanding how natural exposure to sand fly saliva through repeated bites, regardless of *Leishmania* infection, may affect the host response to filaria. Additionally, we also explored how filaria-infected individuals modulate the response to sand fly bites.

As monocytes are recruited to the bite site of disease vectors [42,45], we focused the investigation on understanding the response of these cells in humans that are unexposed or naturally-exposed to sand flies. Interestingly, monocytes recovered from individuals naturally-exposed to sand fly bites were refractory to antigenic challenge, and became unresponsive to filarial or leishmanial antigens, or even potent activators such as LPS. This finding has important implications regarding the response of bite-experienced humans to various pathogens. This is also of particular relevance to *Leishmania* parasites that are transmitted by sand flies since monocytes are their host cells. The effect of monocyte dysfunction upon repeated exposure to sand fly bites on the establishment and progression of *Leishmania* infection remains to be assessed.

In contrast to saliva-exposed monocytes, cells from healthy volunteers produced significantly higher levels of several inflammatory and regulatory cytokines/chemokines in response to filarial and leishmanial antigens but only upregulated the production of CCL4 (MIP-1 β) and IL-6 in response to sand fly saliva. Our findings support previous studies that reported the induction of IL-6 and CCL2, another monocyte chemoattractant, in response to sand fly saliva

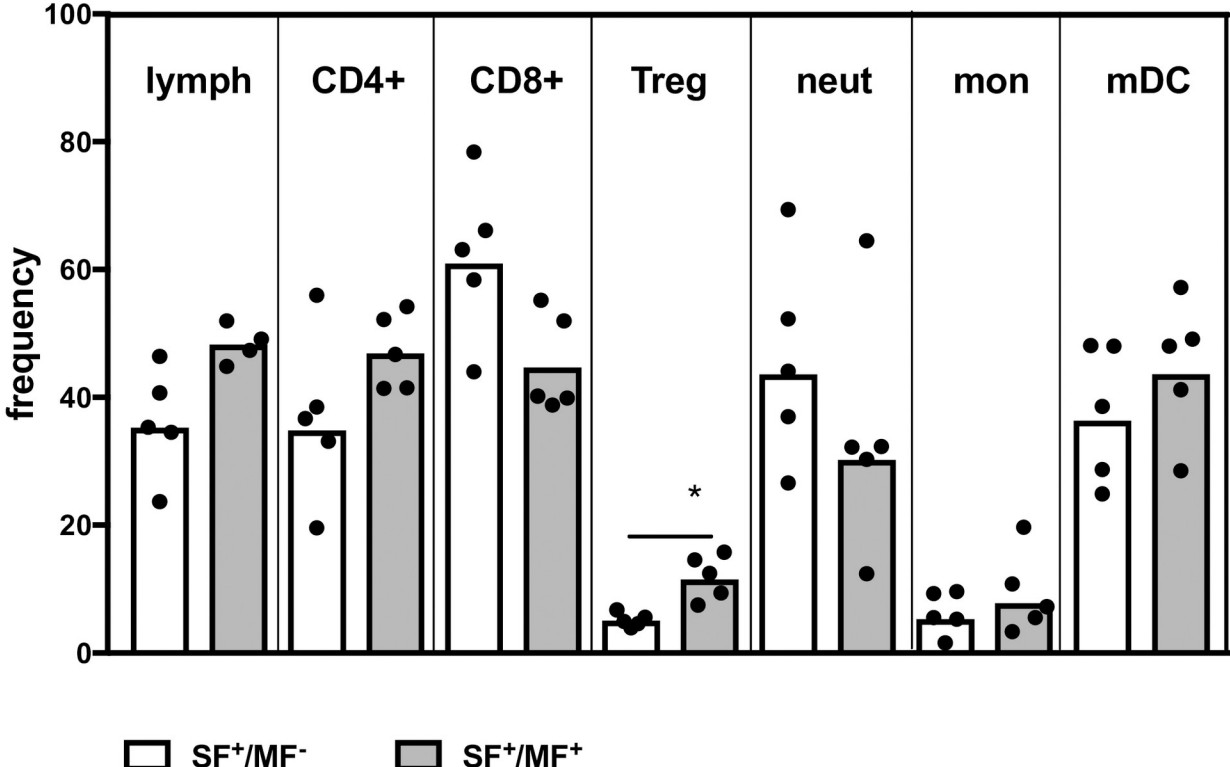

**Fig 5. SF+/MF+ individuals have a higher frequency of Treg than SF+/MF- individuals.** Whole blood from SF+/MF+ (n = 5) and SF+/MF- (n = 5) individuals were RBC-lysed and frequency of different subsets were measured by flow cytometry. Frequency of B lymphocytes were not different between SF+/MF- and SF+/MF+ (1.81±1.21 and 3.96±2.38 respectively, p = 0.11). Statistical difference in frequency of subsets between the two groups are measured using nonparametric Mann- Whitney; * ($p$<0.05).

or bites [42,44]. IL-6 is a proinflammatory cytokine that has been shown to down modulate the killing of *Leishmania amazonensis* in human macrophages [47]. In addition, it has been shown that *Lutzomyia longipalpis* sand fly salivary gland homogenates enhance IL-6 production by human monocytes which may enhance the survival of *Leishmania* in humans infected with the parasite [48]. A study reporting the upregulation of CCL4 by *Leishmania donovani* hypothesized that this parasite induces the expression of CCL4 in host cells to attract regulatory CCR5+ cells to the site of infection, and to further downregulate host immune response [49]. Interestingly, similar to SG Pd, Leish ag, and LPS also upregulate CCL4 in unexposed monocytes, but not if monocytes have been pre-exposed to sand fly bites. This data suggests that repeated exposure to sand fly bites in endemic regions alters the innate immune response rendering them less responsive to further activation with either sand fly salivary proteins, *Leishmania* antigens or LPS. It is important to note that monocytes from Malian individuals produce a higher basal level of cytokines compared to controls, rendering them less responsive to antigen-specific stimulation. However, since the majority of individuals living in endemic areas in Mali have been exposed to sand fly bites, we compared the response of their monocytes to cells from North American individuals as controls. As such, factors other than repeated exposure to sand fly saliva may have contributed to the induction of the heightened state of activation observed for Malian monocytes.

Chronic infection with filarial parasites is known to bias the human immune response towards a more regulatory Th2/M2 profile, hence diminishing the Th1/M1 response [43,50]. In this study we have observed that compared to cells from SF+/MF- subjects, monocytes from

SF$^+$/MF$^+$ individuals produce less cytokines and are more of the M2-phenotype, evidenced by a higher CCL22 mRNA expression and a higher frequency of CCL22$^+$ cells. Furthermore, the frequency of IL-10$^+$ myeloid DC and T regulatory cells are increased in SF$^+$/MF$^+$ as compared to SF$^+$/MF$^-$ individuals, indicative of an immune response biased towards a regulatory state. Further MF ag induced the production of type-1, type 2, and regulatory cytokines in monocytes from individuals not previously exposed to sand fly bites. However, in individuals that are exposed to sand fly bites in endemic regions, microfilariae (SF$^+$/MF$^+$) does not alter basal cytokine production from either monocytes or whole blood. This is in accordance with the helminth induction of both regulatory and type-2 phenotypes [51] which seems not to be affected by exposure to sand fly bites.

This study demonstrates that repeated exposure to sand fly saliva has a profound effect on human monocytes that renders them unresponsive to stimuli. Superimposed on this, the immune response of individuals residing in regions co-endemic for filaria is further modulated by microfilariae, revealing the complexity of the immune interactions that shape the human response to disease vectors and the pathogens they transmit in real-life scenarios. Studies that consider multi-factorial settings that affect the human immune system simultaneously are much needed and may reveal new information vital to understanding how diseases establish in the natural environment.

## Supporting information

**S1 Fig. Short term exposure to saliva of the vector sand fly *Phlebotomus duboscqi* does not alter the mRNA expression of Type 1, Type 2 or regulatory genes in human monocytes.** Human monocytes were unexposed (media) or exposed to either SG Pd, Leish ag, MF ag, LPS, or recombinant proteins 4B4, 1H2, LJ143, or 4C2 for 4 hours. mRNA expression of (A) Type-1 associated genes CCL2, CCL3, CCL4, IP10, and IL-6, (B) Type-2 associated gene CCL22, and (C) regulatory gene IL-10 was measured. Data are represented as box and whiskers (min to max) of 1/average $\Delta C_t$ ($n$ = 5–7). The mRNA detection threshold (dotted line) is 0.027. Statistical differences in mRNA expression are measured using Wilcoxon matched pairs signed-rank test; $^*$ ($p<0.05$).
(TIFF)

**S2 Fig. Basal cytokine levels in overnight monocyte culture of SF$^+$/MF$^-$ and SF$^+$/MF$^+$ individuals.** Monocytes were isolated from PBMC of SF$^+$/MF$^-$ (n = 13) and SF$^+$/MF$^+$ (n = 11) individuals. Cells were cultured in media for 24 hours and supernatant was collected and cytokine levels were measured by Luminex. Bars represent the geometric mean with 95% CI. Statistical analysis was done using nonparametric Mann- Whitney test.
(TIFF)

**S3 Fig. Basal cytokine levels in cultured whole blood of SF$^+$/MF$^-$ and SF$^+$/MF$^+$ individuals.** Whole blood from SF$^+$/MF$^-$ (n = 13) and SF$^+$/MF$^+$ (n = 11) individuals were RBC-lysed and cultured in media for 24 hours. The supernatant was collected and cytokine levels were measured by Luminex. Bars represent the geometric mean with 95% CI.
(TIFF)

**S4 Fig. Anti-Pd SG antibody titers as indicators of sand fly exposure versus basal cytokine expression of all SF$^+$ individuals regardless of MF status.** Our data suggests that there is no significant correlation between any of the tested cytokines and antibody titers. Nevertheless, the cellular immune response may be very different from the antibody response.
(TIFF)

## Acknowledgments

The authors would like to thank the study communities in Kolokani and Kolondieba and the Fieldworkers for their support. We also would like to thank the ICER community and field workers for their dedication and support. We thank Dr. Sasisekhar Bennuru for his assistance with the figures.

## Author Contributions

**Conceptualization:** Moussa Sangare, Yaya Ibrahim Coulibaly, Michel Emmanuel Coulibaly, Siaka Yamoussa Coulibaly, Lamine Soumaoro, Ilo Dicko, Bourama Traore, Sekou Fantamady Traore, Ousmane Faye, Thomas B. Nutman, Jesus G. Valenzuela, Fabiano Oliveira, Seydou Doumbia, Shaden Kamhawi, Roshanak Tolouei Semnani.

**Data curation:** Moussa Sangare, Yaya Ibrahim Coulibaly, Naureen Huda, Silvia Vidal, Sameha Tariq, Michel Emmanuel Coulibaly, Siaka Yamoussa Coulibaly, Lamine Soumaoro, Ilo Dicko, Bourama Traore, Ibrahim Moussa Sissoko, Sekou Fantamady Traore, Ousmane Faye, Thomas B. Nutman, Fabiano Oliveira, Seydou Doumbia, Shaden Kamhawi, Roshanak Tolouei Semnani.

**Formal analysis:** Moussa Sangare, Yaya Ibrahim Coulibaly, Naureen Huda, Silvia Vidal, Sameha Tariq, Michel Emmanuel Coulibaly, Siaka Yamoussa Coulibaly, Lamine Soumaoro, Ilo Dicko, Bourama Traore, Ibrahim Moussa Sissoko, Sekou Fantamady Traore, Ousmane Faye, Thomas B. Nutman, Fabiano Oliveira, Seydou Doumbia, Shaden Kamhawi, Roshanak Tolouei Semnani.

**Funding acquisition:** Moussa Sangare, Yaya Ibrahim Coulibaly, Siaka Yamoussa Coulibaly, Sekou Fantamady Traore, Ousmane Faye, Thomas B. Nutman, Jesus G. Valenzuela, Fabiano Oliveira, Seydou Doumbia, Shaden Kamhawi, Roshanak Tolouei Semnani.

**Investigation:** Moussa Sangare, Yaya Ibrahim Coulibaly, Naureen Huda, Silvia Vidal, Michel Emmanuel Coulibaly, Siaka Yamoussa Coulibaly, Lamine Soumaoro, Ilo Dicko, Bourama Traore, Ibrahim Moussa Sissoko, Sekou Fantamady Traore, Ousmane Faye, Thomas B. Nutman, Fabiano Oliveira, Seydou Doumbia, Shaden Kamhawi, Roshanak Tolouei Semnani.

**Methodology:** Moussa Sangare, Yaya Ibrahim Coulibaly, Naureen Huda, Silvia Vidal, Sameha Tariq, Lamine Soumaoro, Ilo Dicko, Bourama Traore, Ibrahim Moussa Sissoko, Sekou Fantamady Traore, Ousmane Faye, Thomas B. Nutman, Jesus G. Valenzuela, Fabiano Oliveira, Seydou Doumbia, Shaden Kamhawi, Roshanak Tolouei Semnani.

**Project administration:** Moussa Sangare, Yaya Ibrahim Coulibaly, Sekou Fantamady Traore, Ousmane Faye, Thomas B. Nutman, Jesus G. Valenzuela, Fabiano Oliveira, Seydou Doumbia, Shaden Kamhawi, Roshanak Tolouei Semnani.

**Resources:** Moussa Sangare, Ibrahim Moussa Sissoko, Sekou Fantamady Traore, Thomas B. Nutman, Jesus G. Valenzuela, Fabiano Oliveira, Seydou Doumbia, Shaden Kamhawi, Roshanak Tolouei Semnani.

**Software:** Moussa Sangare, Thomas B. Nutman, Fabiano Oliveira, Seydou Doumbia, Shaden Kamhawi, Roshanak Tolouei Semnani.

**Supervision:** Yaya Ibrahim Coulibaly, Thomas B. Nutman, Jesus G. Valenzuela, Fabiano Oliveira, Seydou Doumbia, Shaden Kamhawi, Roshanak Tolouei Semnani.

**Validation:** Silvia Vidal, Thomas B. Nutman, Jesus G. Valenzuela, Fabiano Oliveira, Seydou Doumbia, Shaden Kamhawi, Roshanak Tolouei Semnani.

**Visualization:** Moussa Sangare, Naureen Huda, Silvia Vidal, Thomas B. Nutman, Jesus G. Valenzuela, Fabiano Oliveira, Seydou Doumbia, Shaden Kamhawi, Roshanak Tolouei Semnani.

**Writing – original draft:** Moussa Sangare, Yaya Ibrahim Coulibaly, Naureen Huda, Silvia Vidal, Sameha Tariq, Michel Emmanuel Coulibaly, Siaka Yamoussa Coulibaly, Lamine Soumaoro, Ilo Dicko, Bourama Traore, Ibrahim Moussa Sissoko, Thomas B. Nutman, Fabiano Oliveira, Shaden Kamhawi, Roshanak Tolouei Semnani.

**Writing – review & editing:** Moussa Sangare, Yaya Ibrahim Coulibaly, Naureen Huda, Silvia Vidal, Sameha Tariq, Siaka Yamoussa Coulibaly, Lamine Soumaoro, Sekou Fantamady Traore, Ousmane Faye, Thomas B. Nutman, Jesus G. Valenzuela, Fabiano Oliveira, Seydou Doumbia, Shaden Kamhawi, Roshanak Tolouei Semnani.

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
