## [Decision Letter · Decision Letter 0]

15 Jan 2021

Dear Dr. Sangare,

Thank you very much for submitting your manuscript "Individuals co-exposed to sand fly saliva and filarial parasites exhibit altered monocyte function" for consideration at PLOS Neglected Tropical Diseases. As with all papers reviewed by the journal, your manuscript was reviewed by members of the editorial board and by several independent reviewers. In light of the reviews (below this email), we would like to invite the resubmission of a significantly-revised version that takes into account the reviewers' comments. 

Our sincere apologies for the long delay in this review process, during which ten specialists in the field were successively invited, of which eight unfortunately were unable to review the present manuscript. 

Please address the major comments raised by the reviewers, especially the major limitation of the lack of an endemic SG- group which precludes a firm conclusion about the impact of sand fly pre-exposure.

We cannot make any decision about publication until we have seen the revised manuscript and your response to the reviewers' comments. Your revised manuscript is also likely to be sent to reviewers for further evaluation.

Sincerely,

Johan Van Weyenbergh

Associate Editor

Brian Weiss

Deputy Editor

Reviewer's Responses to Questions

**Key Review Criteria Required for Acceptance?**

**Methods**

-Are the objectives of the study clearly articulated with a clear testable hypothesis stated?

-Is the study design appropriate to address the stated objectives?

-Is the population clearly described and appropriate for the hypothesis being tested?

-Is the sample size sufficient to ensure adequate power to address the hypothesis being tested?

-Were correct statistical analysis used to support conclusions?

-Are there concerns about ethical or regulatory requirements being met?

Reviewer #1: See comments below

Reviewer #2: (No Response)

**Results**

-Does the analysis presented match the analysis plan?

-Are the results clearly and completely presented?

-Are the figures (Tables, Images) of sufficient quality for clarity?

Reviewer #1: See comments below

Reviewer #2: (No Response)

**Conclusions**

-Are the conclusions supported by the data presented?

-Are the limitations of analysis clearly described?

-Do the authors discuss how these data can be helpful to advance our understanding of the topic under study?

-Is public health relevance addressed?

Reviewer #1: See comments below

Reviewer #2: (No Response)

**Editorial and Data Presentation Modifications?**

Reviewer #1: (No Response)

Reviewer #2: (No Response)

**Summary and General Comments**

Reviewer #1: Overall, there seem to be very large differences in basal cytokine/chemokine expression of monocytes from the non-exposed (mainly North American, Fig. 1) versus the endemic individuals (SG+/MF+ and SG+/MF-, Fig. 2). Looking specifically at basal IL-6 and CCL4 levels, these fluctuate between 1 and 200 pg/mL in non-exposed individuals versus 100-20,000 pg/mL in individuals from Mali. This is a 100-fold difference in basal cytokine levels, which is neither discussed nor attempted to be explained in the manuscript. The statement in lines 283-284 is misleading as it only refers to SG+/MF+ and SG+/MF-.

On the background of the very high baseline cytokine levels, I do not agree with the conclusion or concept that the cells would have become unresponsive or dysfunctional due to sand fly exposure. The cells seem to be already strongly activated for one or another reason with only a high LPS concentration still having some limited additional stimulatory effect. 

Moreover, how can authors attribute differences in monocyte responses to prior sand fly exposure per se in the absence of an SG- endemic control group (which could not be identified in the study area as all individuals are SG+).

As the authors have determined the anti-Pd SG antibody titers as indicators of sand fly exposure, could any correlations be found with monocyte responsiveness? This is probably one of the only options to make a link with sand fly exposure.

Where were the Malian individuals recruited?

How was the blood anticoagulated and cryo-preserved? For improved reading flow, monocyte isolation from whole blood (214-2018) may be moved above the section on in vitro culture. After monocyte isolation, how long were monocytes cultured prior to antigenic stimulation. Was this always the same?

The authors indicate in the method section that antigen exposure was for 30min upto 48h (lines 185-186), but only 24h data are reported and a single experiment with a short 40min exposure for a transcriptional study.

The choice of the recombinant salivary proteins, information on their putative functions and references to the methods of production/purification are lacking in the manuscript. Was the Leishmania antigen always from L. major, this is not clear from the methods section and the various figures. Can the authors discuss the choice and potential limitations of Brugia malayi antigen as filarial antigen in the stimulation assays for a study area where other filarial species occur (Wuchereria, Mansonella), i.e. how comparable are the antigenic repertoires.

In figure 1, it is not clear why so few data points were reported for IL-10 as compared to the other cytokines for the SG Pd stimulated condition. Are data points lacking or are they below detection limit? The legend indicates that each line represents an independent donor (n= 6-8). Based on the number of lines, a quick check seems to indicate that there are up to 14 donors in most groups. 

I understand that flow cytometry was done on stimulated/fixed cells of whole blood and on cryopreserved blood. I assume that the same antibody panels were used. Information about the reagents used for intracellular staining seems to be lacking. It is also not entirely clear how the mDCs were identified based on the described panel in lines 223-224. It is clear that additional markers were needed (CD11c+, CD123-, HLA-DR+). For clarity, can authors also indicate the markers used to identify the Tregs and differentiate them from activated T cells. Based on the used panel, I assume that authors can include the B cell frequency as additional information in figure 5 as these also may exert regulatory functions. 

Abstract: the first sentence of the conclusions/significance section seems to fit in the above paragraph. 

Line 165: what is meant by “eluted” blood

IP-10 and the abbreviation for microfilariae is not consistent (capitalization) throughout text and figures.

Greek letters were not readable in figures 4 and supplemental figure 3.

Reviewer #2: (No Response)
---

## [Editor Report · Decision Letter 1]

4 May 2021

Dear Dr. Sangare,

We are pleased to inform you that your manuscript 'Individuals co-exposed to sand fly saliva and filarial parasites exhibit altered monocyte function' has been provisionally accepted for publication in PLOS Neglected Tropical Diseases.

Best regards,

Johan Van Weyenbergh

Associate Editor

Brian Weiss

Deputy Editor

---

## [Editor Report · Acceptance letter]

21 May 2021

Dear Dr. Sangare,

We are delighted to inform you that your manuscript, "Individuals co-exposed to sand fly saliva and filarial parasites exhibit altered monocyte function," has been formally accepted for publication in PLOS Neglected Tropical Diseases.

Best regards,

Shaden Kamhawi

co-Editor-in-Chief

Paul Brindley

co-Editor-in-Chief
